# *Para*-selective nitrobenzene amination lead by C(sp²)-H/N-H oxidative cross-coupling through aminyl radical

Zhen Zhang [1,4] ✉, Shusheng Yue[1,4], Bo Jin[1], Ruchun Yang[1,2], Shengchun Wang [3], Tianqi Zhang[1], Li Sun[1], Aiwen Lei [1,3] ✉ & Hu Cai [1] ✉

Arylamines, serving as crucial building blocks in natural products and finding applications in multifunctional materials, are synthesized on a large scale via an electrophilic nitration/reduction sequence. However, the current methods for aromatic C–H amination have not yet attained the same level of versatility as electrophilic nitration. Here we show an extensively investigated transition metal-free and regioselective strategy for the amination of nitrobenzenes, enabling the synthesis of 4-nitro-*N*-arylamines through C(sp²)-H/N-H cross-coupling between electron-deficient nitroarenes and amines. Mechanistic studies have elucidated that the crucial aspects of these reactions encompass the generation of nitrogen radicals and recombination of nitrobenzene complex radicals. The C(sp²)-N bond formation is demonstrated to be highly effective for primary and secondary arylamines as well as aliphatic amines under mild conditions, exhibiting exceptional tolerance towards diverse functional groups in both nitroarenes and amines (>100 examples with yields up to 96%). Notably, this C(sp²)-H/N-H cross-coupling exhibits exclusive *para*-selectivity.

Arylamines, as structural units in many natural products, are widely applied in the synthesis of medicinal agents, agrochemicals, and multifunctional materials[1–5]. Hence, the construction of C-N bonds to deliver arylamines has become one of the basic transformations in both academia and industry[6]. Traditionally, transition metal-catalyzed C-N cross-coupling have revolutionized the field of synthetic chemistry (Fig. 1a). For several decades, Buchwald-Hartwig[7,8], Chan-Lam couplings[9], and Ullmann amination[10], dominated by Cu- and Pd-based catalytic systems, have revolutionized this area by coupling of amines with aryl halides, pseudohalides or arylboronic acids. Indeed, limitations of these reactions exist, such as the use of pre-functionalized arenes, strong base, and elevated temperatures. In recent years, transition-metal catalysis has also shown great advantage in realizing direct C-H/N-H cross-coupling to facilitate arenes C-H amination without needing preinstallation of a leaving group[11–18].

However, these reactions largely require amines with directing groups and the use of stoichiometric sacrificial oxidants, and the undesired metal by-products remain an environmental concern[19]. Thus, direct amination of C–H bonds without activating or directing groups is considered as atom-economic and operationally efficient transformations. As a result, the pursuing of new strategies for direct C–H amination by C–H/N–H dehydrogenative coupling under mild metal-free conditions is certainly worthy.

Nitroarenes, as versatile building blocks, are obtained from the facile nitration of aromatic compounds and, thus, represent one of easy-to-access nitrogenous partners for the C-N coupling via radical process, which have received significant attention because of the step economy, ready availability, and easy manipulation. In addition, the direct transformation of the NO₂ group has been an attractive option in cross-coupling chemistry[20–24]. Hence, synthetic methods to

[1]School of Chemistry and Chemical Engineering, Nanchang University, Nanchang, Jiangxi, People's Republic of China. [2]Institute of Organic Chemistry, Jiangxi Science and Technology Normal University, Nanchang, Jiangxi, People's Republic of China. [3]College of Chemistry and Molecular Sciences, Wuhan University, Wuhan, Hubei, People's Republic of China. [4]These authors contributed equally: Zhen Zhang, Shusheng Yue. ✉e-mail: zhangzhen@ncu.edu.cn; aiwenlei@whu.edu.cn; caihu@ncu.edu.cn

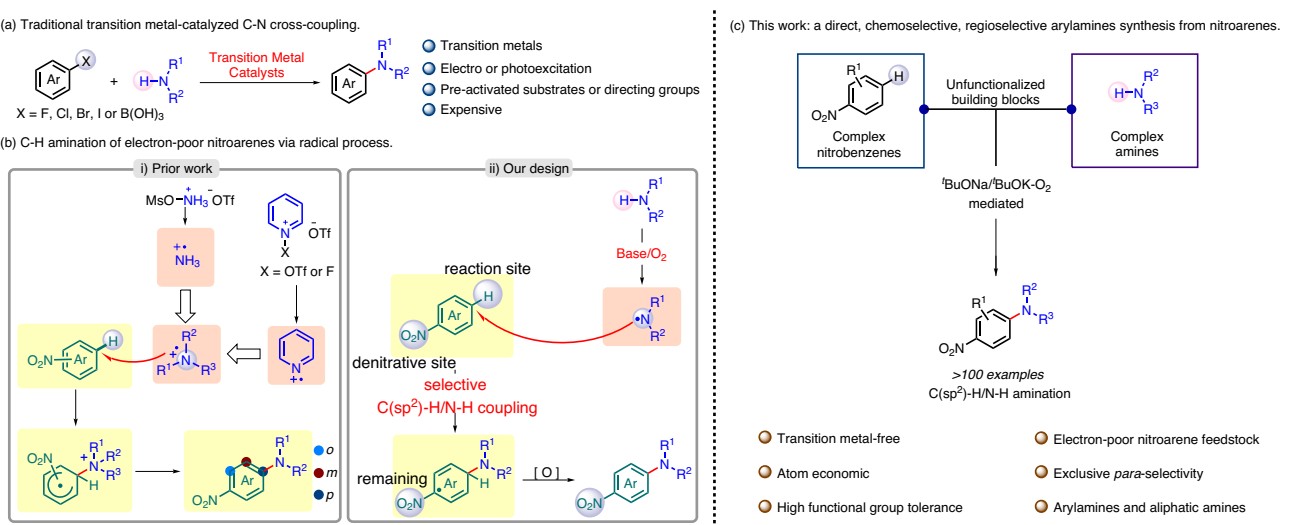

**Fig. 1 | Amination of aromatics and our reaction design. a** Traditional transition metal-catalyzed C-N cross-coupling. **b** Prior work and our design: C-H amination of electron-poor nitroarenes via radical process. **c** This work: a transition metal-free, base/O₂ mediated *para*-selective C-H/N-H amination of nitroarenes with complex amines.

access privileged scaffold arylamines directly from nitroarenes are valuable.

In general, C-H amination of nitroarenes has shown uncommon through the formation of $\sigma^H$ adducts by $S_NArH$ reaction, and simultaneously often afforded products with limited regioselectivity or predominant *ortho*-selectivity[25–28]. The development of new efficient strategies that rely on different functionalized coupling partners, particularly electron-poor nitroarenes, could enable access to synthetically challenging arylamines. More recently, direct radical C−H amination strategies have exhibited particular innovation[29–35], but challenges remain with C−H amination of electron-poor nitroarenes due to the essence of the electron-deficient nitrogen radical. In 2019, Ritter[36,37] and Carreira[38] independently reported the discovery of N-centered radical cations, which offer a direct pathway for the C-H amination of nitroarenes (Fig. 1b). However, polarity matching of radicals and arenes leads to a mixture of positional selectivity, resulting in the formation of *ortho*-, *meta*-, and *para*-substituted products. Thus, the approaches, realizing the straightforward site-selective synthesis of functionalized arylamines by using synthetically upstream nitroarenes with further efficient denitrative transformations of the NO₂ group, are challenging but desirable[20]. For offering new options in step- and atom-economic organic synthesis, we expect that the direct C−H amination could be achieved directly from nitroarenes and simple amines for the C(sp²)-N coupling.

Herein, we report a synthetic protocol leading to the efficient and highly regioselective synthesis of 4-nitro-*N*-arylamines, which have been widely found in optoelectronic fields[39], advanced materials[40], and starting materials of conducting functional polymers[41] (Fig. 1c). Under mild transition metal-free conditions, selective C(sp²)-H/N-H cross-coupling is realized through nitrogen radicals of complex arylamines and especially aliphatic amines. Compared with arylamines, alkylamines with the enhanced instability of their corresponding nitrogen radicals, often show much more challenging related reactivities and tend to translate into highly stabilized α-N carbon-radicals[33].

## Results
### Optimization and Scope Elucidation
Initially, 1,2,3,4-tetrahydroquinoline **1a-1** and nitrobenzene **2a** were chosen as model substrates to investigate suitable conditions for the regioselective *para*-site cross-coupling. We then try to optimize the reaction conditions, and it turned out that the reaction was observed to proceed only in DMSO and DMF among the investigated solvents. To our delight, the reaction could deliver the expected coupling product **3a-1** in 71% yield under O₂ with DMSO as solvent and *ᵗ*BuONa as base (Supplementary Table 1). Next, we found that indoline derivatives **1b** also reacted well with nitrobenzene **2a** to give exclusively the *para*-substituted products **3b**. We began with the reaction of indoline **1b-1** and **2a** for the optimization of reaction. Solvents and bases screening showed that DMSO and *ᵗ*BuOK were still the best solvent/base system for this transformation to give **3b-1** in 52 % yield. However, we found that when DMF was used as solvent and *ᵗ*BuOK (4.0 equiv.) as base, lowering the reaction temperature to −30°C could enhance the yield of **3b-1** to 85% (Supplementary Table 10).

With the established optimum conditions in hand, we explored the substrate scope of amines derivatives **1**. The reaction exhibited compatibility with a range of 1,2,3,4-tetrahydroquinoline derivatives **1a** and indoline derivatives **1b** as summarized in Fig. 2. Consequently, nitrobenzene C-H amination products **3a-1** to **3b-24** were obtained with moderate to high yields (30-94%) and exclusive *para*-regioselectivity. The aryl rings of **1a** and **1b** exhibited excellent tolerance towards various substituents, including electron-donating groups (-Me, -OMe, -O*ⁱ*Pr, -OH) and electron-withdrawing groups (-F, -Cl, -Br, -COOH), resulting in the anticipated products. The presence of a methyl group at C2-C6 (**3b-2**-**3b-8**) led to a slight decrease in yield, particularly when positioned at C3 (45%, **3b-3**) and C5 (58%, **3b-5**). Figure 2 demonstrates the inability of **1a-26** with substituents at C8 to yield the product. However, the reaction between indole derivative 7-methylindoline **1b-24**, bearing a methyl group at C7, and compound **2a** resulted in the formation of product **3b-24** with a low yield of 30%. The ester group in **1a-25** underwent hydrolysis to form the carboxyl group under alkaline conditions, leading to the synthesis of product **3a-23** with a yield of 79%. Furthermore, a moderate yield of 54% was also achieved for the formation of **3b-23** from indole **1b-23**. The relative configuration of **3a-6** and **3b-1** was unambiguously assigned by X-ray analysis of single crystal.

In an effort to broaden the substrate scope for amines, we are delighted to observe that primary amines such as 1- and 2-naphthylamines (**1c-1** and **1c-2**), acyclic *N*-methylanilines with various substituents (-Me, -OMe, -halogen, -ethynyl, -1*H*-pyrrol-1-yl, -CF₃, -OCF₃, -SMe, -carbonyl) on the phenyl ring as well as *N*-methylpyridin-2-amine **1c-9** and diphenylamine **1c-10** can also participate in reactions with **2a** to yield *para*-nitroaryl amines **3c-1**-**3c-19** in high yields (Fig. 2c). However, the reactions involving other arylprimary amines yielded unsatisfactory results, with the prominent formation of azobenzene compounds observed as ones of defined by-products. The reaction proceeds

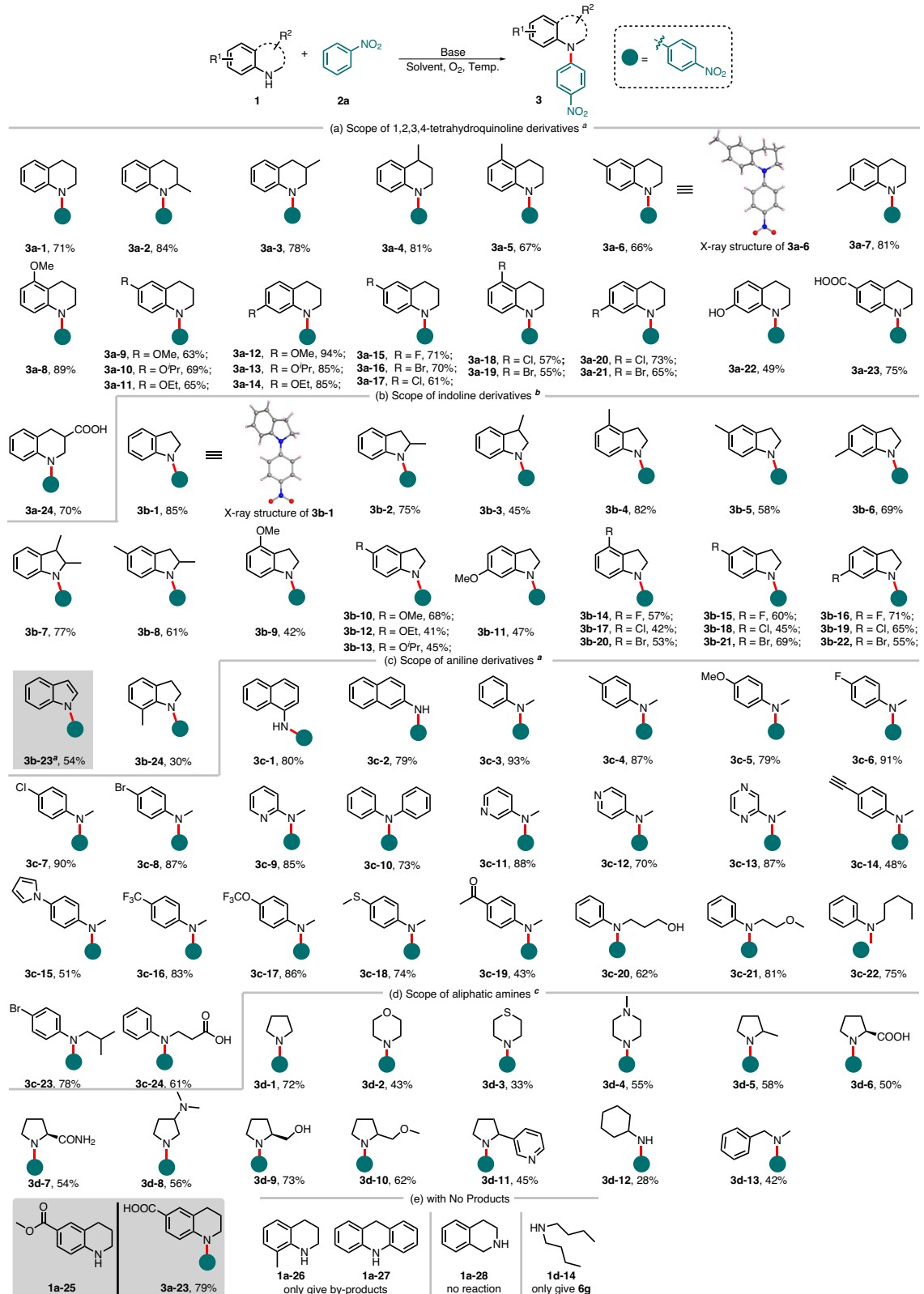

**Fig. 2 | Substrate scope of 1 with 2a. a** Scope of 1,2,3,4-tetrahydroquinoline derivatives **1a**. Reaction conditions A: **1** (0.5 mmol), **2a** (4.0 equiv.), tBuONa (3.0 equiv.), DMSO (3.0 mL), O₂ (1.0 atm) and the reaction was conducted at 40 °C. **b** Scope of indoline derivatives **1b**. Reaction conditions B: **1** (0.5 mmol), **2a** (3.0 equiv.), tBuOK (4.0 equiv.), DMF (5.0 mL), O₂ (1.0 atm) and the reaction was conducted at −30 °C. **c** Scope of aniline derivatives **1c**. **d** Scope of aliphatic amines **1d**. Reaction conditions C: **1** (0.2 mmol), **2a** (4.0 equiv.), tBuOK (3.0 equiv.), DMSO (3.0 mL), O₂ (1.0 atm) and the reaction was conducted at 40 °C. **e** Substrates with no products. Isolated yield.

effectively even when the methyl group of *N*-methylaniline is substituted with various aliphatic chains or functional groups (**3c-20**-**3c-24**).

The successful reaction of various simple aliphatic amines, such as secondary amines pyrrolidines **3d-5**-**3d-11** with multiple functional groups, morpholine **3d-2**, thiolmorpholine **3d-3** and 1-methylpiperazine **3d-4**, primary amine cyclohexylamine **3d-2**, as well as *N*-methyl-1-phenylmethanamine **3d-13**, highlights their potential for achieving desired products. The unexpected product 4-nitro-*N*-phenylaniline **6 g** was obtained with a yield of less than 20%. It appears that the formation of **6 g** can be attributed to the reduction of **2a** to aniline **5d** within the system, followed by its subsequent reaction with **2a**.

Unfortunately, certain amines failed to yield the anticipated products. Even when subjected to reactions in DMF at −50 °C and −10 °C respectively, 8-methyl-1,2,3,4-tetrahydroquinoline **1a-26** and 9*H*-carbazole **1a-27** only underwent oxidative decomposition. The reaction did not proceed from 1,2,3,4-tetrahydroisoquinoline **1a-28** and even under elevated temperatures exclusively afforded isoquinoline as the product via oxidative dehydrogenation. To our surprise, we didn't get the product from dibutylamine **1d-14** but only **6 g** with 15% yield.

Nitroarenes, as privileged scaffold of chemical synthesis, are of great significance in organic synthesis. Thus, the substrate scope of nitrobenzene derivatives **2** with amines **1** was explored. Under mild conditions, the reaction has delivered diverse arylamines **4a-1**-**4a-20** in moderate to high yields (30-96%), showing high functional group tolerance (Fig. 3). Exclusive *para*-selective regioselectivity was achieved from various electron-donating (-Me, -OMe, -SMe) and electron-withdrawing groups (-F, -Cl, -Br, -CN, -NO$_2$, -CF$_3$, -CONH$_2$, -COOMe), either at *ortho*- or *meta*-position of the phenyl rings. It is interesting to see that we could obtain the products **4a-3** and **4b-3** in moderate yields from the reaction of 3-nitrobenzonitrile with the cyclic amines (**1a-1** and **1b-1**), while similar reaction with **1c-1** only gave the desired product **4c-3** in 30% yield at −30 °C. Product **4b-10** was not formed from **1b-10** and 1,2-dinitrobenzene **2k**, while **4a-10** and **4c-10** were observed in moderate yields. The boric acid group was eliminated from nitroboric acid **2a-19** during the reaction, leading to the formation of product **3a-1**, while the ester group underwent alkaline hydrolysis, resulting in the formation of a hydroxyl group (**4a-20**). Furthermore, the reaction is incompatible with functional groups such as ketone, alcohol, and carboxylic acid. The structures of **4a-7**, **4b-8**, **4b-9** and **4c-10** were unambiguously confirmed by single-crystal X-ray analysis.

**Mechanistic studies.** Although a few nitroarene *o*- and *p*- aminations have been reported and were suggested to proceed by S$_N$Ar$^H$ via the nucleophilic attack of the aminyl anion to thenitroarene, the mechanisms of these reactions should be studied in depth[42,43]. Although formally resembling an S$_N$Ar$^H$ reaction, control experiments and radical clock experiments demonstrate that the reaction proceeds via a radical mechanism in the DMSO/'BuONa/O$_2$ or DMF/'BuOK/O$_2$ system. Further prove of the radical mechanism comes from the reaction of 4-phenyl-1,2,3,4-tetrahydroquinoline **5a** with **2a**, which gives the 1,4-di(4-nitrophenyl) substituted product **6a** in 41% yield (Fig. 4a). Similar reaction of 3-methyl-4-phenyl tetrahydroquinoline **5b** with **2a**, on the other hand, affords only the product **6b** in 75% yield, probably manifesting the sensibility of a radical reaction toward steric hindrance (Fig. 4b). At the same time, the reaction of 4-methyltetrahydroquinoline **5c** with **2a** could not proceed to give the 4-nitrophenyl product **6c** (Fig. 4c). These facts clearly indicate that in the formation of product **6a**, the process of deprotonation of the benzylic C-H bond followed by attack of the resulting carbanion to **2a** is not involved. Meanwhile, a thermodynamic consideration showed that a proton transfer from the 4-benzylic C-H bond (p$K_a$ ~ 33 in DMSO) to tetrahydroquinolinyl anion (p$K_a$ ~ 29.5 in DMSO)[44] is not feasible. These results strongly disfavor a pathway involving nucleophilic attack by an aminyl anion

but can be rationalized by a radical mechanism (Fig. 4d). Deprotonation/oxidation of **5a** yielded the aminyl radical **6a-I**, where thermodynamically favorable intramolecular 1,4-hydrogen atom shift from the 4-benzylic C-H bond (Bond dissociation energy (BDE) ~ 73 kcal/mol) to the 4-aminyl radical (BDE of C-H bond ~ 89 kcal/mol) furnished the 4-benzylyc radical **6a-II**[45]. Recombination of **6a-II** with **2a** gave the primary product **6a-III**, whose further similarly reaction with **2a** led to the final product **6a**. The reaction didn't almost happen under N$_2$ atmosphere after deoxygenation by lyophilization for five times (Fig. 4e, f). The reaction can give azobenzene **6d** with aniline **5d** from radical homo-coupling under standard conditions, indicating the nitrogen radical (Fig. 4g)[46,47].

Considering the analogous characteristics shared by benzonitrile **5f** and **2a**, we subsequently investigated the reactivity of **5f** with **1a-1**. However, no product was obtained even under elevated temperatures (Fig. 4h). The comparison of critical data between their respective transition state intermediates is imperative for elucidating the role of the NO$_2$ group. The NO$_2$ group demonstrates a superior ability to stabilize negative charge in comparison to the CN group during C-N bond formation. Besides, both Mayer Bond Order (MBO) and electron density ρ of C...N bond in **TS1** of **1a-1** surpass those in **TS1c** of **5f**, indicating that **TS1** has stronger C...N bond than **TS1c**, thereby accounting for the lower observed free energy in **TS1** (Supplementary Fig. 12). It is noteworthy that this reaction is independent of visible light and can occur even under dark conditions (Fig. 4i). Moreover, revealing results have demonstrated that the specific alkali metal ions exert a substantial influence on reactions involving aliphatic amines. The reaction did not proceed as expected under standard conditions, resulting in the formation of an unexpected product **6e** from **1d-1** (Fig. 4j). Upon modifying the standard conditions, another unexpected product **6 g** was obtained in addition to **6e**. It appears that the conversion of **2a** into aniline **5d** occurred, followed by a subsequent reaction between **5d** and **2a** to yield the desired *para*-substituted nitrobenzene C-H amination product **6 g** (Fig. 4k). However, when 'BuOK was used as the base instead of 'BuONa (Fig. 4l), the reaction proceeded successfully to afford product **3d-1** along with compound **6g**. To further validate the reaction mechanism, a radical clock experiment was conducted using *N*-cyclopropyl-3-methoxyaniline **5i** and **2a** to confirm the plausibility of a radical-mediated pathway (Fig. 4m). The expected product **6i-1** was determined by GC-MS in the system, resulting in the formation of **6i** with a yield of 35%. We postulated that **5i** could undergo SET to generate radical **5i-I**, which subsequently decomposed into aniline **5j**. The resulting aniline then engaged in a reaction with **2a** to afford the desired product **6i**. To elucidate the mechanistic pathway, control experiments were carried out. According to these experiment results, we further came to the conclusion that the reaction proceeds through a radical pathway (Supplementary Fig. 2).

To further investigate the reaction mechanism, confirmation experiments are designed by Electron Paramagnetic Resonance (EPR), further confirming a radical pathway (Fig. 5). When there wasn't **2a**, another competing reaction is that nitrogen radical tended to be easily transformed into aminoxyl radical (g = 2.0054, A$_N$ = 11.5 G) in the DMSO/'BuONa/O$_2$ system through Korcek's radical-trapping antioxidant (RTA)[48], and was also obvious after reacting for 10 min. Furthermore, the absence of 'BuONa or O$_2$ resulted in no observation of any nitrogen radical, indicating that both the base and O$_2$ are essential for the formation of nitrogen radical.

For further understanding of the reaction mechanism, density functional theory (DFT) was calculated (Fig. 6). As mentioned above, the reaction gave azobenzene **6d** with aniline **5d** from radical homo-coupling under standard conditions, indicating the nitrogen radical (Fig. 4g). We then took **1a-1** as an example to illustrate the possible mechanism of corresponding *N*-radicals (denoted as **1a-1-radical**). Firstly, **1a-1** was combined with 'BuONa to form complex **int1(1a-1)**,

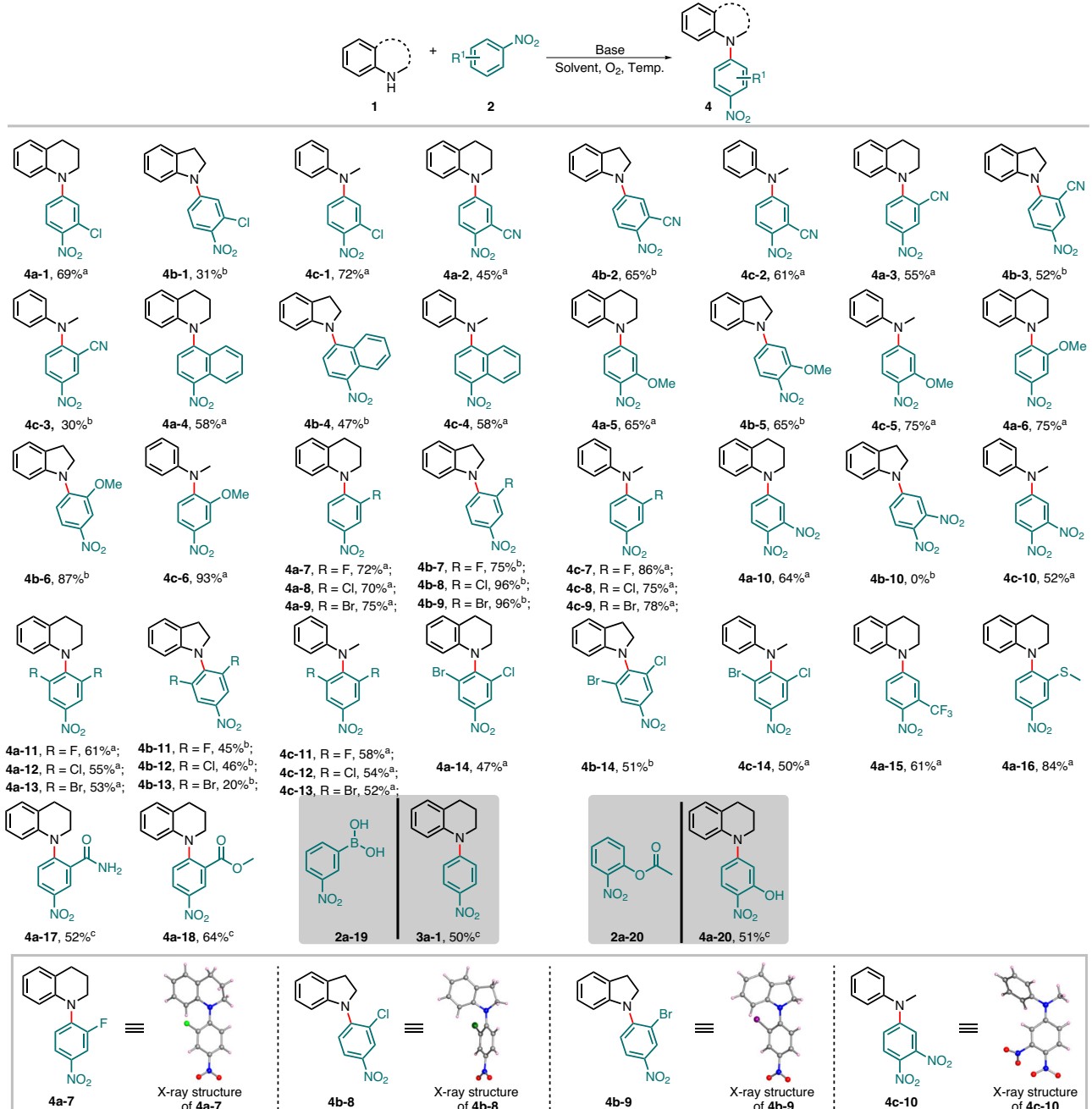

**Fig. 3 | Substrate scope of nitrobenzene derivatives 2 [a,b,c,d].** [a] Reaction conditions A: **1** (0.5 mmol), **2** (4.0 equiv.), *t*BuONa (3.0 equiv.), DMSO (3.0 mL), O₂ (1.0 atm) and the reaction was conducted at 40 °C. [b] Reaction conditions B: **1** (0.5 mmol), **2** (3.0 equiv.), *t*BuOK (4.0 equiv.), DMF (5.0 mL), O₂ (1.0 atm) and the reaction was conducted at -30 °C. [c] Reaction conditions C: **1a-1** (0.2 mmol), **2** (4.0 equiv.), *t*BuONa (6.0 equiv.), DMSO (3.0 mL), O₂ (1.0 atm) and the reaction was conducted at 40 °C. [d] Isolated yield.

then proton transfer occurred via **Ts1(1a-1)** to give **int2(1a-1)**. It should be noticed that $G$(**Ts1(1a-1)**) is slightly lower than $G$(**int2(1a-1)**) but $E$(**Ts1(1a-1)**) is higher than $E$(**int2(1a-1)**), indicating this step is barrier-less (Fig. 6a). Afterwards *t*BuOH was released to generate **int3(1a-1)**, and O₂(triplet) then oxidized **int3(1a-1)** into **int4(1a-1)(triplet)**, which is a diradical.

Finally, **1a-1-radical** was obtained by releasing Na⁺O₂⁻. It can be seen that the formation of **1a-1-radical** is only 7.3 kcal/mol endothermic, and the reaction free energy barrier is quite low. In the *t*BuONa/DMSO/O₂ system, **1a-1-radical** can attack both *para* and *ortho* positions of NO₂ of **2a**, resulting two transition states **TS1** and **TS1′** correspondingly. Δ$G$(**TS1**) is lower than Δ$G$(**TS1′**), indicating that *para*

position attack is more preferred, resulting the **INT1**. Next step is the aromatization reaction of **INT1**, and Na⁺O₂⁻ can abstract the H atom via **TS2**, resulting the **INT2**(triplet) state. Thus, **INT2 (triplet)** was converted into **INT2 (singlet)**, and then released both **NaO₂H** and **3a-1**. After all, a possible mechanism is proposed (Fig. 6b).

**Practical application.** Triphenylamine derivatives, as organic electroluminescent materials, are of great potential for various optoelectronic applications[49–52]. However, their practical applications are still hampered by lack of efficient and convenient synthetic methods. The approach, realizing the straightforward synthesis of functionalized arylamines by using synthetically upstream nitroarenes, offers

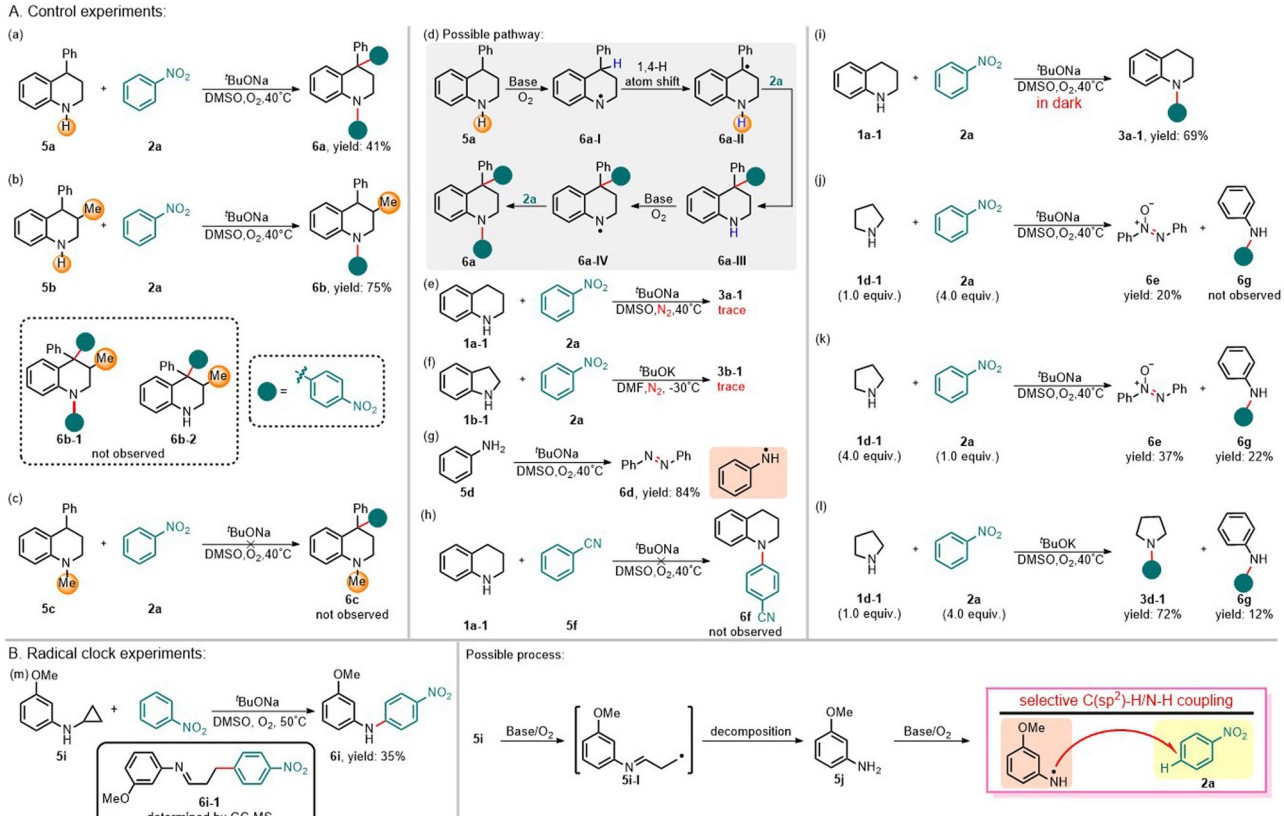

**Fig. 4 | Mechanistic studies. A** Control experiments. **a**–**d** The essential role played by the N-H bonds of **1**. **e**, **f** The crucial role of O₂ in the reactions. **g** Radical homo-coupling of **5d**. **h** The substitution of the nitro group with cyanide is not feasible. **i** Experiment in dark. **j**–**l** The impact of reactant stoichiometry and base type on the reaction. **B** Radical clock experiments. **m** The radical clock experiment of **5i** with **2a** and the possible process.

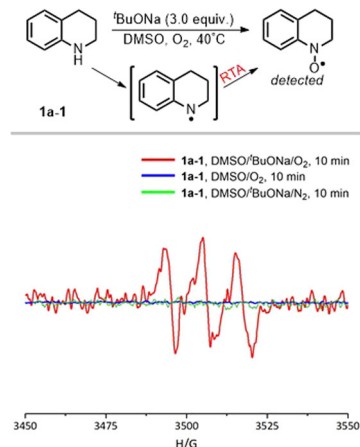

**Fig. 5 | EPR studies.** Reaction conditions: **1a-1** (0.5 mmol), ᵗBuONa (3.0 equiv.), O₂ (1.0 atm), react in DMSO (3.0 mL) for 10 min.

powerful synthetic options to construct diverse chemical bonds often found in, for example, advanced materials and pharmaceuticals by further efficient denitrative transformations of the NO₂ group in cross-coupling chemistry[20–24]. With the feasibility of practical application of the approach, we then evaluated the scalability of the reaction by performing a gram-scale reaction. To our delight, the scaled-up reaction kept the high reaction efficiency and gave **3c-10** (1.20 g) from **1c-10** (1.20 g, 7.1 mmol) in 70% yield (Fig. 7a). Especially, the direct utilization of the synthetically upstream nitroarenes leads to step- and atom-economic access to the formation of C-C, C-N, and C-H bonds (Fig. 7b).

## Discussion

In summary, we have successfully developed a direct C-H and N-H dehydrogenative coupling reaction between amines and nitroaromatic compounds in DMSO/ᵗBuONa/O₂ or DMF/ᵗBuOK/O₂ system to provide an efficient and versatile synthetic method for *para*-nitroarylamine derivatives. The reaction proceeds by aminyl radicals coupling mechanism with the environmentally benign O₂ as an oxidant under mild and transition metal-free conditions. Further advantages include the good functional tolerance and wide substrate scope in regard to both amines and nitroarenes, and exclusive *para*-regioselectivity. Mechanistic studies have demonstrated that in the DMSO/ᵗBuONa(ᵗBuOK)/O₂ system, O₂ (triplet) could deliver nitrogen radicals as an oxidant from Na-amide, thereby providing a mild and versatile approach to accessing synthetically significant N-centered radicals. Meanwhile, electron acceptor compounds such as nitroarenes are prone to transform into anion radicals in this system. These may open new reaction pathways via radical-radical anion recombination. Further applications of this reaction system are underway in our laboratories.

## Methods

### Reaction conditions A

A clean, oven-dried Schlenk tube with previously placed magnetic stirbar was charged with 1,2,3,4-tetrahydroquinoline (66.6 mg, 0.5 mmol), nitrobenzene (246.3 mg, 2.0 mmol) and ᵗBuONa (144.2 mg, 1.5 mmol) in dry DMSO (3 mL) solvent under argon atmosphere. The reaction was evacuated and back filled with O₂ (1.0 atm) and this sequence was repeated for three additional times. The reaction mixture was vigorously stirred at 40°C and monitored by TLC. After the complete consumption of 1,2,3,4-tetrahydroquinoline, the reaction mixture was cooled to room temperature and then quenched with water (5 mL),

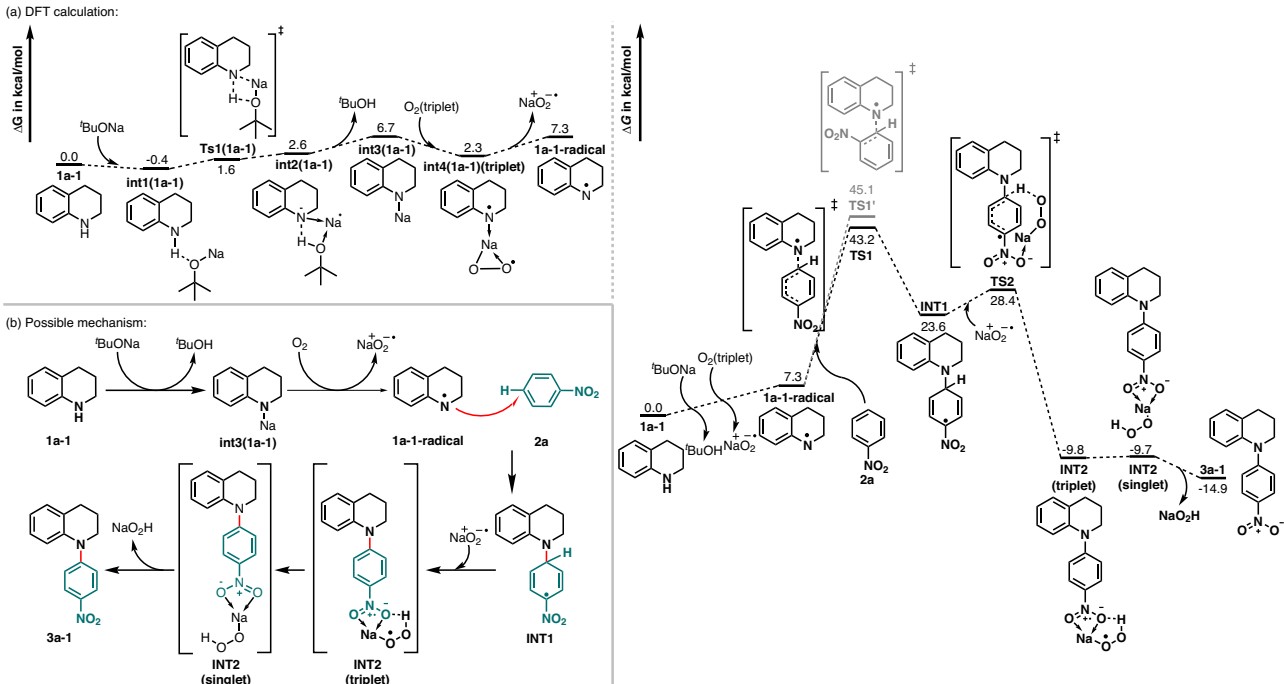

**Fig. 6 | Computational studies. a** Relative Gibbs free energies (in kcal·mol$^{-1}$): The free energies of **1a-1, 2a**, O$_2$ (triplet), and $^t$BuONa were set to 0.0 kcal·mol$^{-1}$ as a reference. **b** Proposed reaction mechanism.

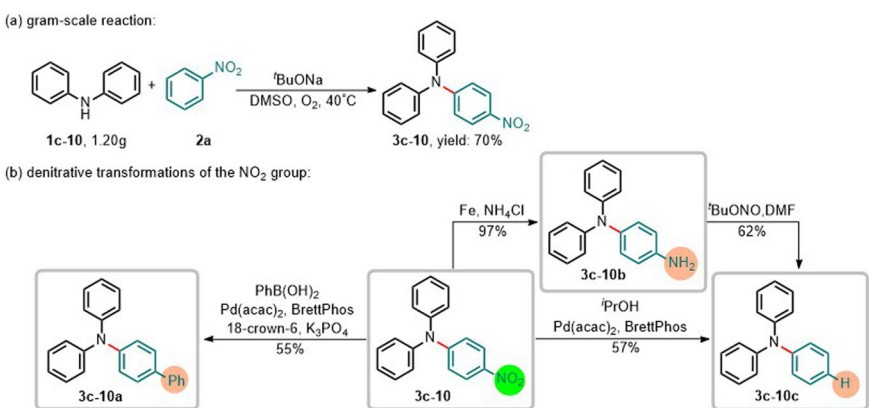

**Fig. 7 | Synthetic applications. a** Scaled-up reaction. **b** Denitrative transformations of the NO$_2$ group.

diluted with ethyl acetate, and extracted with ethyl acetate (25 mL × 3). The combined organic phases were washed with brine (5 mL), dried over Na$_2$SO$_4$ and concentrated in vacuo. The residue was purified by neutral Al$_2$O$_3$ column chromatography (PE: EtOAc = 50:1, v/v).

### Reaction conditions B

A clean, oven-dried Schlenk tube with previously placed magnetic stir-bar was charged with nitrobenzene (184.7 mg, 1.5 mmol), $^t$BuOK (224.5 mg, 2.0 mmol) in dry DMF (5 mL) solvent at room temperature under argon atmosphere. After the reaction mixture was stirred at −30 °C for 10 min, the reaction was evacuated and back filled with O$_2$ (1.0 atm) and this sequence was repeated for three additional times. After indoline (59.6 mg, 0.5 mmol) was added, the reaction stirred at −30 °C and monitored by TLC. After the complete consumption of indoline, the reaction mixture was quenched with water (5 mL), diluted with ethyl acetate, and extracted with ethyl acetate (25 mL × 3). The combined organic phases were washed with brine (5 mL), dried over Na$_2$SO$_4$ and concentrated in vacuo. The residue was purified by neutral Al$_2$O$_3$ column chromatography (PE: EtOAc = 40:1, v/v).

### Reaction conditions C

A clean, oven-dried Schlenk tube with previously placed magnetic stir-bar was charged with pyrrolidine (27.3 mg, 0.3 mmol), $^t$BuOK (109.8 mg, 0.90 mmol), nitrobenzene (147.6 mg, 1.20 mmol) in dry DMSO (3 mL) solvent at room temperature under argon atmosphere. The reaction was evacuated and back filled with O$_2$ (1.0 atm) and this sequence was repeated for three additional times. The reaction mixture was vigorously stirred at 40 °C and monitored by TLC. After complete conversion of pyrrolidine, the reaction mixture was restored to room temperature and then quenched with water. diluted with ethyl acetate, and extracted with ethyl acetate (25 mL × 3). The combined organic phases were washed with saturated NaCl aqueous solution. The combined organic phases were washed with brine (5 mL), dried over Na$_2$SO$_4$, and concentrated in vacuo. The residue was purified by silica gel column chromatography (PE: EtOAc = 80:1, v/v).

## Data availability

The X-ray crystallographic coordinates for structures reported in this study have been deposited at the Cambridge Crystallographic Data

Centre (CCDC), under deposition numbers 2105132, 2105448, 2176176, 2176175, 2175172. These data can be obtained free of charge from The Cambridge Crystallographic Data Centre via www.ccdc.cam.ac.uk/data_request/cif. Crystallographic data for the structures reported in this article have been deposited at the Cambridge Crystallographic Data Centre, under deposition numbers CCDC 2105132 (**3a-6**), CCDC 2105448 (**3b-1**), CCDC 2176176 (**4a-7**), CCDC 2176175 (**4b-8**), CCDC 2175172 (**4b-9**) and CCDC 2181802 (**4c-10**). Copies of the data can be obtained free of charge via https://www.ccdc.cam.ac.uk/structures/. The data generated or analyzed during this study are included in this article and the supplementary information. The Cartesian coordinates are available from the Source Data. Details about materials and methods, experimental procedures, characterization data, computational details, and NMR spectra are available in the Supplementary Information. All data are available from the corresponding author upon request. Source data are provided with this paper.

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

## Acknowledgements

We are grateful for financial support from the National Natural Science Foundation of China (21861026, H.C. and 22075123, H.C.). Especially, we are very grateful to Prof. Dr. Jianhua Xu at the Institute of Chemistry and Chemical Engineering, Nanjing University, for his suggestions on the reaction mechanism.

## Author contributions

Z.Z. designed and carried out the experiments, and did the theoretical calculations, and analyzed the data and wrote the manuscript. S.Y. carried out the experiments under the supervision of Z.Z. and H.C. B.J., R.Y., T.Z., and L.S. carried out part of the experiments. S.W. analyzed the EPR data. A.L. and H.C. supervised the project.

## Competing interests

The authors declare no competing interests.
