## [Peer Review File · Nature Communications]

Para-Selective Nitrobenzene Amination Lead by C(sp²)-H/N-H
Oxidative Cross-Coupling through Aminyl RadicalREVIEWER COMMENTS

Reviewer #1 (Remarks to the Author):

[See attached file]

Reviewer #2 (Remarks to the Author):

The article by Cai & Lei describes a method for synthesizing para-nitroaniline through the oxidative coupling of an amino radical with nitroarene. This process requires only potassium or sodium tert-butoxide and an oxygen atmosphere (1 ATM) as the oxidant, along with a protic solvent at a moderate range of temperatures. The authors have reported a broad substrate scope (more than 100 molecules), although redundant in terms of structural diversity and functional groups. The reaction has been scaled to gram quantities, indicating its potential for industrial applications. Mechanistic insights to prove the radical nature of the reaction have been provided through EPR spectroscopy. While the research introduces a completely novel mechanism, it is intrinsically bound to a limited class of molecules, namely nitroaniline. This limitation is not significant given the importance of this compound class. However, the diversity on the amine side is lacking, demonstrating versatility mainly in just three types of amines: 1,2,3,4-tetrahydroquinoline, indoline, and methyl aniline. I believe that adding diversity would strongly support the case for publication and would greatly enhance the rapid adoption of the reported chemistry.

I would be happy to consider the paper for publication in Nature Communication after major revisions where the following points are addressed in order to prove the synthetic impact of the protocol and enhance the manuscript's quality, particularly in the introduction and the discussion of the mechanism.

1. The paper needs to be checked by native English speaker because many parts are hard to understand.
2. The introduction is broad and well-written, but I think the discussion about Professor Baran's work doesn't fit with this paper and might confuse readers. This paper talks about making a C-N bond between the ring's Sp² carbon and an amine. In Professor Baran's paper, anilines are made by a different process, which is a formal alkylation of the nitro group. I suggest removing it.
3. The authors should reference and discuss three significant studies in the introduction that describe the CH amination of electron-poor arenes, including nitrobenzene, through radical chemistry: Ritter et al. in *Chemical Science*, 2019, 10, pages 2424-2428; Ritter et al. in *Angewandte Chemie International Edition*, 2019, 58, pages 532-536; and Carreira et al. in *Angewandte Chemie International Edition*, 2019, 58, pages 526-531. I am surprised that this paper are completely omitted considering the relevance with the reported reaction.
4. The authors should address the synthesis of nitroaniline through SnAr on Sp² C-H bonds. This is important because these kinds of reactions are rare but use very similar conditions.(ex: *J. Org. Chem.* 1998, 63, 4878-4888, *Chem. Commun.*, 2016, 52, 7237-7240 , *Synthesis* 2010, 22, 3865–3872, *Journal of the Chemical Society* 1932, 1254-1261)
5. When addressing the research of Professor Ritter and Professor Leonori, the authors

should revise their statement."More recently, direct radical amination strategies have showed particular innovation, but they are usually applied to electron-rich aromatics due to the essence of the electron-deficient nitrogen radical" to electron-rich and electroneutral, for example Fluorobenzene is effectively aminated in these works together with other electron neutral arene.

6. Citation 30 is off-topic.

7. The authors have reported a considerable number of amines, yet the scope is notably redundant. Mainly, 1,2,3,4-tetrahydroquinoline and indoline are featured, and the substitution pattern on the aromatic ring is limited, with only halogens, alkyl ether, and methyl group. Could the authors provide at least 5 examples where other functional groups are incorporated onto the ring?

It would be beneficial to see if functional groups such as cyanide, ester, ketone, trifluoromethyl, boronic ester, azide, thioether, sulfone, and amide are compatible with the reaction.

8. Methyl phenyl alanine is well represented, but in a repetitive manner. Could the author provide 5 additional examples where the methyl group is replaced with different aliphatic chains or groups.

9. In the abstract, the authors state that they have achieved coupling with primary arylamines and aliphatic amines, but only two examples of each are provided in the scope. Could the author provide 5 additional examples for each of these categories (5+5), as they are important building blocks and while really demonstrate the generality of the protocol.

10. Do primary aliphatic amines not react? If that is the case, can the authors specify this as a limitation of the protocol in the main text.

11. Regarding the aromatic scope, the authors tested three categories of amine for each nitroarene, but the number of nitroarenes used is very limited, with only 14 examples. I would prefer to see reactions of just 1,2,3,4-tetrahydroquinoline, but with more diversity among the nitroaromatics. Can the authors include at least another 5 to 8 different nitroarenes in the scope? It would be interesting to see if functional groups such as cyanide, ester, ketone, trifluoromethyl, boronic ester, azide, thioether, sulfone, and amide are compatible with the reaction.

12. What happens when para-nitrotoluene and parafluorobenzene react? Can the authors report the outcome of such a reaction? Does the selectivity change, or does the reaction not work? Could the author provide this information?

13. Would the author consider rephrasing this sentence for clarity?

"It formally seems to be an S_NArH reaction, however, as shown in our optimization experiments that the reaction proceeds only in the DMSO/tBuONa/O₂ or DMF/tBuOK/O₂ system, which is characterized by the presence of alkali metal ion as a possible mediator of further reactions via single electron transfer (SET) from a reactant to generate the free radicals of the latter.

14. Could the authors please provide references to substantiate the claim that tBuONa or tBuOK serves as a mediator in reactions proceeding via single electron transfer, as

mentioned in lines 161-162-163

15. Can the author provide a proper citation to support the statement made at lines 182-183. "The reaction can give azobenzene 6d with aniline 5d from radical homo-coupling under standard conditions, indicating the nitrogen radical"

16. Could the authors repeat the experiment in fig 2g for synthesizing azobenzenes with methyl aniline, as reported in the manuscript, since there are no examples using simple aniline.

17. Could you please elucidate how the experiments shown in Figures 2h, 2j, 2k, and 2l relate to the mechanism under discussion?

Presenting these experiments without deriving conclusions may lead to confusion. It may be more appropriate to relocate this information to another section or include it in the Supplementary Information. Particularly the reaction in Figure 2h appears to have an energy profile (S40 in the SI) that closely resembles that of nitrobenzene could the author provide an explanation?

18. The Radical clock experiment reported in Fig 2m did not provide any product that prove radical formation. Furthermore, the explanation offered for the formation of 6g lack experimental support.

-Could the authors clarify how they have ruled out the possibility that a simple reduction of nitroarene to form aniline, followed by the formation of compound 6g as described earlier, is not occurring?

-It may be beneficial for the authors to consider repeating the experiment with a different nitroarene, such as ortho-Cl or ortho-F, or paramethyl N-cyclopropylaniline to ensure that a self-condensation reaction to form 6g as described earlier is not taking place.

19. The experiment reported on page S35 of the SI, Figure 2B, which show the mass detection of the opened radical clock is quite significant and therefore should be included in the main manuscript, since provides tentative support for the formation of the aminyl radical. To substantiate this finding, the authors are encouraged to provide the FID)from the mass spectrometry, including fragmentation patterns, in the SI.

20. Would the author consider rephrasing this sentence for clarity? Line 205-207 Which radical species is the author referring to?

"Furthermore, nitrogen radical could not deliver the desired radical without tBuONa or O₂, indicating that the base and O₂ are all necessary for the formation of nitrogen radical."

21. At lines 207-208, the text mentions 'the obvious nitrobenzene complex radical.' Is this radical the same as the one shown in EPR Figure 3b and described as INT1 in Figure 4? For clarity, the should call it with is name and repor the structure in Figure 3b next to EPR spectra.

22. Furthermore, the details about the EPR on page S33, which provide the most convincing evidence of the radical nature of the process, should be included in the main manuscript.

23. Regarding the conclusion, the authors suggest that triplet oxygen could generate nitrogen radicals through hydrogen abstraction from the N-H bond, presenting a mild and versatile pathway to N-centered radicals. However, the mechanism illustrated in Figure 4 depicts oxygen functioning merely as an oxidant for Na-amide, with triplet oxygen acting as a

simple oxidant. Could the Authors clarify this point?

24. In the Supplementary Information, the FID, ¹H and ¹³C for compounds 1b-11, 1b-12, 1b-13, 1b-16, 1b-17, 4b-4, and 4c-11 show impurities in the aromatic region. These compounds should be further purified, and higher-quality data should be provided

Reviewer #3 (Remarks to the Author):

This manuscript by Lei and Cai reports a para-selective nitrobenzene amination. This reaction was carried out through dehydrogenative C-H/N-H cross-coupling between nitroarenes and amines in DMSO/base/O₂ system through aminyl radical process. This is an interesting finding. The mechanistic studies were well designed. Considering its interesting finding, I recommend this manuscript to be published in Nat Commun after the minor revision.

- 1) In Figure 1b, the radical position in Baran's work should be corrected.
- 2) In Figure 2m, the reaction of 5g with 2a gave 6g. However, 6g was also possible to be obtained from homo-coupling of 2a (see Fig. 2l). A substituted 5g is recommended for the reaction to verify the aniline moiety of 6g being from 5g.
- 3) How about if the para-position of nitroarenes is blocked. Can the authors provide some examples?

Reviewer #4 (Remarks to the Author):

In this manuscript, Cai and Lei et al. reported a strategy to synthesize 4-nitro-N-arylamines by C(sp²)-H/N-H oxidative cross-coupling reaction between amines and nitroaromatic compounds in DMSO/t-BuOK/O₂ system. Under mild and transition-metal-free conditions, para-nitroarylamines derivatives could be efficiently obtained by this versatile synthetic method using O₂ as oxidant. The mechanistic studies show that the reaction is a free radical reaction, rather than a nucleophilic attack of the amino anion to the nitroarene. This manuscript could be accepted after considering the following issues:

1. From the substrate scope in Table 1 and Table 2, the synthesis strategy is only applicable to secondary amines instead of primary amines. The range of substrate is should be corrected in the abstract. In addition, the synthesis strategy seems to rely strongly on the conjugation of aryl groups with amines. In the experiment with only four groups of aliphatic amines as substrates (3d-1, 3d-2, 1a-24, 1d-3), only tetrahydrofuran got the expected products. It can be seen that this strategy is also not suitable for aliphatic amines.
2. The substrate scope in Table 1 and Table 2 only contain some simple substituent groups (alkoxy, halogen and methyl groups). The authors could try more representative substrate. Meanwhile, I wonder if the strategy be functional group tolerance or not? For example, how about the tolerance for the carboxylic acid, alcohol, amide, aldehyde, ketone and other common functional groups?
3. Table 1 shows that 1,2,3,4-tetrahydroquinoline derivatives with substituents at C8 could not obtain products. Similarly, could the indole derivatives with substituents at C7 have reactivity?
4. Could pyrrole be used as substrate in this strategy?
5. In this work, does the nitrobenzene with substituents at C4 position react or not? And what

is the role of the nitro group? Could it be replaced by other electron-withdrawing groups?

6. It is no doubt that the reaction is a free radical reaction proved by the control experiment. However, the DFT calculation is confused. The energy barrier of the reaction reached 43.2 kcal/mol from 1a-1 to TS1, and the reaction could not happen under the experimental conditions. In addition, the generation of NaO₂ species is suspicious. Is it a free radical? Could it be verified by experiments?

The manuscript reports on the development of a procedure for site-selective *para*-C(sp²)-H amination of nitrobenzene enabled by the DMSO(DMF)/*t*BuOK(Na)/O₂ systems.

I am sincerely amazed by the very low quality of this contribution. The manuscript is poorly written, careful editing of the entire text should have been carried out prior to submission. Moreover, the mechanistic part is confused and highly speculative, and does not provide any conclusive evidence in favor of the proposed mechanism. The manuscript cannot be accepted for publication in Nature Communications. In the present form, I would also definitely not recommend publication in a more specialized synthetic journal.

Abstract. The authors state that ...”A transition metal-free and regioselective nitrobenzene amination strategy to synthesize 4-nitro-*N*-arylamines through dehydrogenative C(sp²)-H/N-H cross-coupling between electron-poor nitroarenes and amines in DMSO/*t*BuOK/O₂ system is well established.”

Because this is actually the outcome of the present work I think that the concept of well established is not appropriate.

Figure 2. Panels (b) ii and (c) are very similar and should be merged into a single one.

Page 3, Results. The authors state that...”the reaction could not proceed in solvents other than DMSO and DMF.”

Additional details of the different solvents employed must be provided. The sentence should be probably reformulated taking into account the solvents that have been tested and could read as follows: among the investigated solvents, the reaction was observed to proceed only in DMSO and DMF.

Page 3, lines 106 and 114. What do the authors mean with relative configuration of **3a-6** and **3b-1**?

Page 3, lines 109-111. The substituents are typically not allowed to give similar yields. Please replace with a more pertinent definition.

Page 4, lines 128-129. Product **6g** should be displayed in Table 1 and properly quantified.

Page 4, line 132. In order to help the reader, the mentioned by-products should be clearly defined.

The description of the mechanistic studied that starts at the end of page 5 is confused and often speculative. Lines 160-163: ...”the reaction proceeds only in the DMSO/*t*BuONa/O₂ or DMF/*t*BuOK/O₂ system, which is characterized by the presence of alkali metal ion as a possible mediator of further reactions via single electron transfer (SET) from a reactant to generate the free radicals of the latter.”

Please elaborate, as it stands the sentence is not clear

Line 166: tetrahydroquinoline

Lines 175-178. The authors speculate that...”Deprotonation/oxidation of **5a** yielded the aminyl radical **6a-I**, where thermodynamically favorable intramolecular 1,4-hydrogen atom shift from the 4-benzylic C-H bond (Bond dissociation energy (BDE) ~ 73 kcal/mol) to the 4-aminyl radical (BDE of C-H bond ~ 89 kcal/mol) furnished the 4-benzylic radical **6a-II**.”

Although thermodynamically favourable, I sincerely don't see why and especially how the proposed intramolecular 1,4-hydrogen atom shift occurs

Lines 192-199. The description of the radical clock experiment is pure speculation. *N*-Cyclopropylaniline is not an appropriate probe for the intended purpose. The putative intermediate aminyl radical is stabilized by the phenyl group and this would significantly slow down ring opening. The possible process described on the bottom-left of Figure 2 makes no sense. Product **6g** is observed in a number of reactions where no aromatic

amine is employed and this reviewer does not understand the need to propose a tentative mechanism for the formation of this product from N-cyclopropylaniline without any experimental support.

The authors should indicate if the DFT calculations refer to gas-phase or if a solvent model was employed

Page 9. Discussion should be replaced by Conclusions

P.S.: The below please find our revisions and all the responses to the reviewers' comments.

Response to reviewer 1's comments

Comment & Suggestions 1.1:

Abstract. The authors state that "A transition metal-free and regioselective nitrobenzene amination strategy to synthesize 4-nitro-*N*-arylamines through dehydrogenative C(sp²)-H/N-H cross-coupling between electron-poor nitroarenes and amines in DMSO/^tBuOK/O₂ system is well established." Because this is actually the outcome of the present work, I think that the concept of well-established is not appropriate.

Our Replies:

The phrase "...well established" has been replaced with "...well explored".

Comment & Suggestions 1.2:

Figure 2. Panels (b)-ii and (c) are very similar and should be merged into a single one.

Our Replies:

The introduction of the design details in Panels (b)-ii and an overview of the reaction in Panels (c) would be more effective if presented separately.

Comment & Suggestions 1.3:

Page 3, Results. The authors state that ... "the reaction could not proceed in solvents other than DMSO and DMF." Additional details of the different solvents employed must be provided. The sentence should be probably reformulated taking into account the solvents that have been tested and could read as follows: among the investigated solvents, the reaction was observed to proceed only in DMSO and DMF.

Our Replies:

The different solvents employed have been previously described in detail prior to this latest release. Please refer to ESI, †S17.

Comment & Suggestions 1.4:

Page 3, lines 106 and 114. What do the authors mean with relative configuration of **3a-6** and **3b-1**?

Our Replies:

The phrases have been replaced with "The structure of **3b-1** was unambiguously

confirmed by single-crystal X-ray analysis” and “The structures of **4a-7**, **4b-8**, **4b-9** and **4c-10** were unambiguously confirmed by single-crystal X-ray analysis”.

Comment & Suggestions 1.4:

Page 3, lines 109-111. The substituents are typically not allowed to give similar yields. Please replace with a more pertinent definition.

Our Replies:

Revised in this section, please refer to lines 93-105.

Comment & Suggestions 1.5:

Page 4, lines 128-129. Product **6g** should be displayed in Table 1 and properly quantified.

Our Replies:

The formation of **6g** is an unexpected outcome observed in radical-clock experiments. Additionally, the utilization of a substituted **5i** compound is attempted to confirm the presence of the aniline moiety in **6i** originating from **5i**. And the expected product **6i-1** was determined by GC-MS in the system. Unfortunately, our attempts to isolate the target product **6i-1** were unsuccessful.

Comment & Suggestions 1.6:

Page 4, line 132. In order to help the reader, the mentioned by-products should be clearly defined.

Our Replies:

The compounds 8-methyl-1,2,3,4-tetrahydroquinoline **1a-22** and 9H-carbazole **1a-23** underwent oxidative decomposition. The exact identification of the resulting by-products remains challenging.

Comment & Suggestions 1.7:

The description of the mechanistic studied that starts at the end of page 5 is confused and often speculative. Lines 160-163: ... “the reaction proceeds only in the

DMSO/^tBuONa/O₂ or DMF/^tBuOK/O₂ system, which is characterized by the presence of alkali metal ion as a possible mediator of further reactions via single electron transfer (SET) from a reactant to generate the free radicals of the latter.” Please elaborate, as it stands the sentence is not clear.

Our Replies:

The phrase has been replaced with “Although formally resembling an S_NArH reaction, control experiments and radical clock experiments demonstrate that the reaction proceeds via a radical mechanism in the DMSO/^tBuONa/O₂ or DMF/^tBuOK/O₂ system”.

Comment & Suggestions 1.8:

Line 166: tetrahydroquinoline.

Our Replies:

The word “tetrahydroquiniline” has been replaced with “tetrahydroquinoline”.

Comment & Suggestions 1.9:

Lines 175-178. The authors speculate that ... “Deprotonation/oxidation of **5a** yielded the aminyl radical **6a-I**, where thermodynamically favorable intramolecular 1,4-hydrogen atom shift from the 4-benzylic C-H bond (Bond dissociation energy (BDE) ~ 73 kcal/mol) to the 4-aminylic radical (BDE of C-H bond ~ 89 kcal/mol) furnished the 4-benzylic radical **6a-II**.” Although thermodynamically favourable, I sincerely don’t see why and especially how the proposed intramolecular 1,4-hydrogen atom shift occurs.

Our Replies:

To gain mechanistic insight into this ^tBuOK/DMSO/O₂ process, we investigated the influence of N-H bond in the reaction (Fig. 2a and 2c). Among control experiments, substrate **5a** gave the 1,4-di(4-nitrophenyl) substituted product **6a** with a moderate yield of 41%, while substrate **5c** containing an *N*-substituent (*N*-Me) didn't react with **2a**. These results underscore the essential role played by the N-H bond and suggest potential involvement of intermediates such as **6a-I** and **6a-II**.

Control experiments:

Comment & Suggestions 1.10:

Lines 192-199. The description of the radical clock experiment is pure speculation. *N*-Cyclopropylaniline is not an appropriate probe for the intended purpose. The putative intermediate aminyl radical is stabilized by the phenyl group and this would significantly slow down ring opening. The possible process described on the bottom-left of Figure 2 makes no sense. Product **6g** is observed in a number of reactions where no aromatic amine is employed and this reviewer does not understand the need to propose a tentative mechanism for the formation of this product from *N*-cyclopropylaniline without any experimental support.

Our Replies:

The utilization of a substitute **5i** is attempted to confirm the presence of the aniline moiety in **6i** originating from **5i**. And the expected product **6i-1** was determined by GC-MS in the system. Unfortunately, our attempts to isolate the target product **6i-1** were unsuccessful.

Comment & Suggestions 1.11:

The authors should indicate if the DFT calculations refer to gas-phase or if a solvent model was employed

Our Replies:

We optimized structures in gas-phase, and calculated single-point energies in SMD solvation model (DMSO as solvent).

Comment & Suggestions 1.12:

Page 9. Discussion should be replaced by Conclusions.

Our Replies:

“Discussion” has been replaced by “Conclusions”.

Response to reviewer 2' s comments

Comment & Suggestions 2.1:

The paper needs to be checked by native English speaker because many parts are hard to understand.

Our Replies:

We have carefully revised our writing, and corrected grammatically incorrect sentences in our manuscript.

Comment & Suggestions 2.2:

The introduction is broad and well-written, but I think the discussion about Professor Baran's work doesn't fit with this paper and might confuse readers. This paper talks about making a C-N bond between the ring's Sp^2 carbon and an amine. In Professor Baran's paper, anilines are made by a different process, which is a formal alkylation of the nitro group. I suggest removing it.

Our Replies:

The discussion pertaining to Professor Baran's work has been omitted.

Comment & Suggestions 2.3:

The authors should reference and discuss three significant studies in the introduction that describe the C-H amination of electron-poor arenes, including nitrobenzene, through radical chemistry: Ritter et al. in *Chemical Science*, **2019**, *10*, pages 2424-2428; Ritter et al. in *Angewandte Chemie International Edition*, **2019**, *58*, pages 532-536; and Carreira et al. in *Angewandte Chemie International Edition*, **2019**, *58*, pages 526-531. I am surprised that this paper are completely omitted considering the relevance with the reported reaction.

Our Replies:

In the introduction, we have referenced these studies as 36, 37, and 38 while discussing the radical-based C-H amination of electron-deficient arenes (lines 61-64).

Comment & Suggestions 2.4:

The authors should address the synthesis of nitroaniline through S_NAr on Sp^2 C-H bonds. This is important because these kinds of reactions are rare but use very similar conditions (ex: *J. Org. Chem.* **1998**, *63*, 4878-4888, *Chem. Commun.* **2016**, *52*, 7237-7240, *Synthesis* **2010**, *22*, 3865–3872, *Journal of the Chemical Society* **1932**, 1254-1261).

Our Replies:

Lines 54-56, we have addressed the aforementioned synthesis of nitroaniline via S_NAr on $C(sp^2)$ -H bonds.

Comment & Suggestions 2.5:

When addressing the research of Professor Ritter and Professor Leonori, the authors should revise their statement. “More recently, direct radical amination strategies have showed particular innovation, but they are usually applied to electron-rich aromatics due to the essence of the electron-deficient nitrogen radical” to electron-rich and electroneutral, for example Fluorobenzene is effectively aminated in these works together with other electron neutral arene.

Our Replies:

Lines 58-61, the phrase has been replaced with “More recently, direct radical C–H amination strategies have exhibited particular innovation, but challenges remain with C–H amination of electron-poor nitroarenes due to the essence of the electron-deficient nitrogen radical”.

Comment & Suggestions 2.6:

Citation 30 is off-topic.

Our Replies:

The reference 30 has been removed.

Comment & Suggestions 2.7:

The authors have reported a considerable number of amines, yet the scope is notably redundant. Mainly, 1,2,3,4-tetrahydroquinoline and indoline are featured, and the substitution pattern on the aromatic ring is limited, with only halogens, alkyl ether, and

methyl group. Could the authors provide at least 5 examples where other functional groups are incorporated onto the ring?

It would be beneficial to see if functional groups such as cyanide, ester, ketone, trifluoromethyl, boronic ester, azide, thioether, sulfone, and amide are compatible with the reaction.

Our Replies:

We have provided 10 additional examples. The ester group in **1a-25** underwent hydrolysis to form the carboxyl group under alkaline conditions, leading to the synthesis of product **3a-23** with a yield of 79%.

Comment & Suggestions 2.8:

Methyl phenyl alanine is well represented, but in a repetitive manner. Could the author provide 5 additional examples where the methyl group is replaced with different aliphatic chains or groups.

Our Replies:

We have provided 5 additional examples where the methyl group is replaced with different aliphatic chains or groups.

Comment & Suggestions 2.9:

In the abstract, the authors state that they have achieved coupling with primary arylamines and aliphatic amines, but only two examples of each are provided in the scope. Could the author provide 5 additional examples for each of these categories (5+5), as they are important building blocks and while really demonstrate the generality of the protocol.

Our Replies:

We have provided 11 additional examples of aliphatic amines. However, the reactions involving other arylprimary amines yielded unsatisfactory results, with the prominent formation of azobenzene compounds observed as ones of defined by-products. The limitations of this type of substrates have been delineated within the main manuscript (lines 113-115).

Comment & Suggestions 2.10:

Do primary aliphatic amines not react? If that is the case, can the authors specify this as a limitation of the protocol in the main text.

Our Replies:

As mentioned above, primary aliphatic amine 1-cyclohexylamine **1d-12** did react with **2a** to afford **3d-12** with a low yield of 28%.

Comment & Suggestions 2.11:

Regarding the aromatic scope, the authors tested three categories of amine for each nitroarene, but the number of nitroarenes used is very limited, with only 14 examples. I would prefer to see reactions of just 1,2,3,4-tetrahydroquinoline, but with more diversity among the nitroaromatics. Can the authors include at least another 5 to 8 different nitroarenes in the scope? It would be interesting to see if functional groups such as cyanide, ester, ketone, trifluoromethyl, boronic ester, azide, thioether, sulfone, and amide are compatible with the reaction.

Our Replies:

We have provided 6 different functional groups (-CN, -CF₃, -SMe, -CONH₂, -COOMe, -OCOMe) of nitroarenes. The boric acid group was eliminated from nitroboronic acid **2a-19** during the reaction, leading to the formation of product **3a-1**, while the ester group underwent alkaline hydrolysis, resulting in the formation of a hydroxyl group (**4a-20**).

Comment & Suggestions 2.12:

What happens when *para*-nitrotoluene and *para*-fluorobenzene react? Can the authors report the outcome of such a reaction? Does the selectivity change, or does the reaction not work? Could the author provide this information?

Our Replies:

Using nitrobenzene with substituents at C4 position, substrates **2a-1** and **2a-1** only gave the product **3a-1**, but substrate **2a-1** didn't react with **1a-1**.

Comment & Suggestions 2.13:

Would the author consider rephrasing this sentence for clarity? “It formally seems to be an S_NArH reaction, however, as shown in our optimization experiments that the reaction proceeds only in the DMSO/^tBuONa/O₂ or DMF/^tBuOK/O₂ system, which is characterized by the presence of alkali metal ion as a possible mediator of further reactions via single electron transfer (SET) from a reactant to generate the free radicals of the latter.”

Our Replies:

The phrase has been replaced with “Although formally resembling an S_NArH reaction, control experiments and radical clock experiments demonstrate that the reaction proceeds via a radical mechanism in the DMSO/^tBuONa/O₂ or DMF/^tBuOK/O₂ system”.

Comment & Suggestions 2.14:

Could the authors please provide references to substantiate the claim that ^tBuONa or ^tBuOK serves as a mediator in reactions proceeding via single electron transfer, as mentioned in lines 161-162-163.

Our Replies:

In accordance with suggestion **2.13**, our ideas were expressed imprecisely and subsequently rephrased.

Comment & Suggestions 2.15:

Can the author provide a proper citation to support the statement made at lines 182-183. “The reaction can give azobenzene **6d** with aniline **5d** from radical homo-coupling under standard conditions, indicating the nitrogen radical”

Our Replies:

Please refer to reference 46 (*Science* **2008**, *312*, 1806–1809) and reference 47 (*ACS Catal.* **2013**, *3*, 478–486).

Comment & Suggestions 2.16:

Could the authors repeat the experiment in Fig 2g for synthesizing azobenzenes with methyl aniline, as reported in the manuscript, since there are no examples using simple aniline.

Our Replies:

The experiment in Fig 2g was repeated to synthesize azobenzene using methyl aniline **1c-1**, however, instead of azobenzene, (*E*)-*N*-methyl-*N*,*N'*-diphenylformimidamide **6j** was only obtained as the product with a low yield of 15%. It is worth noting that examples include simple anilines such as **1c-1** and **1c-2**.

Comment & Suggestions 2.17:

Could you please elucidate how the experiments shown in Figures **2h**, **2j**, **2k**, and **2l** relate to the mechanism under discussion?

Presenting these experiments without deriving conclusions may lead to confusion. It may be more appropriate to relocate this information to another section or include it in the Supplementary Information. Particularly the reaction in Figure **2h** appears to have an energy profile (S40 in the SI) that closely resembles that of nitrobenzene could the author provide an explanation?

Our Replies:

Based on the results presented in Figures 2j, 2k, and 2l, it is evident that the type of alkali metal ions exerts a significant influence on reactions involving aliphatic amines.

The comparison of critical data between **TS1** and **TS1c** in Figures 2h, respectively, is essential for investigating the role of the NO₂ group (Figure SX1).

During C-N bond formation, in **PhNO₂→TS1** procedure, the negative charge of NO₂ group ($q(\text{NO}_2)$) is increased from -0.193 to -0.256, and the $q(\text{NO}_2)$ change $\Delta q(\text{NO}_2) = -0.063$. While in **PhCN→TS1c** procedure, the negative charge of CN group ($q(\text{CN})$) is only increased from -0.206 to -0.228, and the $q(\text{CN})$ charge $\Delta q(\text{CN}) = -0.022$. The NO₂ group can stabilize more negative charge than CN group during C-N bond formation. Besides, both Mayer Bond Order (MBO) and electron density ρ of C...N bond in **TS1** are higher than those in **TS1c**, indicating that **TS1** has stronger C...N bond than **TS1c**, which is responsible for lower free energy of **TS1**.

According to the Arrhenius formula, we determined that the relative rates of **TS1** and **TS1c** are in a ratio of 1:0.38, indicating that the rate of **TS1c** was comparatively slower than that of **TS1**. Consequently, the corresponding yield of **5f** was also lower compared to **2a**. It should be noted that due to limitations in computational precision and cost considerations, our results provide qualitative accuracy rather than quantitative precision.

Figure SX1. The results of Hirshfeld charge calculations, Mayer Bond Order (MBO) analyses and QTAIM analyses of **PhNO₂**, **TS1**, **PhCN** and **TS1c**.

Comment & Suggestions 2.18:

The Radical clock experiment reported in Fig. 2m did not provide any product that prove radical formation. Furthermore, the explanation offered for the formation of **6g** lack experimental support.

-Could the authors clarify how they have ruled out the possibility that a simple reduction of nitroarene to form aniline, followed by the formation of compound **6g** as described earlier, is not occurring?

-It may be beneficial for the authors to consider repeating the experiment with a different nitroarene, such as *ortho*-Cl or *ortho*-F, or *paramethyl N*-cyclopropylaniline to ensure that a self-condensation reaction to form **6g** as described earlier is not taking place.

Our Replies:

The utilization of a substitute **5i** is attempted to confirm the presence of the aniline moiety in **6i** originating from **5i**. And the expected product **6i-1** was determined by GC-MS in the system. Unfortunately, our attempts to isolate the target product **6i-1** were unsuccessful.

Comment & Suggestions 2.19:

The experiment reported on page S35 of the SI, Figure 2B, which show the mass detection of the opened radical clock is quite significant and therefore should be included in the main manuscript, since provides tentative support for the formation of the aminyl radical. To substantiate this finding, the authors are encouraged to provide the FID) cfrom the mass spectrometry, including fragmentation patterns, in the SI.

Our Replies:

We did the mass detection of the opened radical clock by GC-MS. After reacting 4.0h, **6i-1**, and **5j** were determined by GC-MS. As the reaction progressed, we found that **6i** increased significantly. Unfortunately, our attempts to isolate the target product **6i-1** were unsuccessful.

4.0h:

18.0h:

Comment & Suggestions 2.20:

Would the author consider rephrasing this sentence for clarity? Line 205-207 Which radical species is the author referring to? “Furthermore, nitrogen radical could not deliver the desired radical without ^tBuONa or O₂, indicating that the base and O₂ are all necessary for the formation of nitrogen radical.”

Our Replies:

As shown in Fig. 3, the absence of ^tBuONa or O₂ resulted in no observation of any nitrogen radical, indicating that both the base and O₂ are essential for the formation of nitrogen radical.

Fig. 3 | EPR studies. Reaction conditions: **1a-1** (0.5 mmol), ^tBuONa (3.0 equiv.), O₂ (1.0 atm), react in DMSO (3.0 mL) for 10 min.

Comment & Suggestions 2.21:

At lines 207-208, the text mentions 'the obvious nitrobenzene complex radical.' Is this radical the same as the one shown in EPR Figure 3b and described as INT1 in Figure 4? For clarity, the should call it with is name and report the structure in Figure 3b next to EPR spectra.

Our Replies:

The radical species previously referred to as “the obvious nitrobenzene complex radical” has been identified as “the nitrobenzene radical anion” ($g = 2.0046$, $A_N = 10.8$ G, $A_H = 3.5$ G, $A_H = 3.5$ G, $A_H = 3.5$ G, $A_H = 0.8$ G, $A_H = 0.8$ G), and subsequently excluded from further discussion.

Comment & Suggestions 2.22:

Furthermore, the details about the EPR on page S33, which provide the most convincing evidence of the radical nature of the process, should be included in the main manuscript.

Our Replies:

The details regarding the EPR have been incorporated into the main manuscript.

Comment & Suggestions 2.23:

Regarding the conclusion, the authors suggest that triplet oxygen could generate nitrogen radicals through hydrogen abstraction from the N-H bond, presenting a mild and versatile pathway to *N*-centered radicals. However, the mechanism illustrated in Figure 4 depicts oxygen functioning merely as an oxidant for Na-amide, with triplet oxygen acting as a simple oxidant. Could the Authors clarify this point?

Our Replies:

Lines 270-272, O₂(triplet) functions as an oxidant, and we have elucidated this aspect in our manuscript.

Comment & Suggestions 2.24:

In the Supplementary Information, the FID, ¹H and ¹³C for compounds **1b-11**, **1b-12**, **1b-13**, **1b-16**, **1b-17**, **4b-4**, and **4c-11** show impurities in the aromatic region. These compounds should be further purified, and higher-quality data should be provided.

Our Replies:

These compounds have undergone further purification, resulting in the provision of higher-quality data in the Supplementary Information.

Response to reviewer 3' s comments

Comment & Suggestions 3.1:

In Figure 1b, the radical position in Baran's work should be corrected.

Our Replies:

The discussion regarding Professor Baran's work has been omitted.

Comment & Suggestions 3.2:

In Figure 2m, the reaction of **5g** with **2a** gave **6g**. However, **6g** was also possible to be obtained from homo-coupling of **2a** (see Fig. 2l). A substituted **5g** is recommended for the reaction to verify the aniline moiety of **6g** being from **5g**.

Our Replies:

The utilization of a substituted **5i** is attempted to confirm the presence of the aniline moiety in **6i** originating from **5i**. And the expected product **6i-1** was determined by GC-MS in the system. Unfortunately, our attempts to isolate the target product **6i-1** were unsuccessful.

Comment & Suggestions 3.3:

How about if the *para*-position of nitroarenes is blocked. Can the authors provide some examples?

Our Replies:

Using nitrobenzene with substituents at C4 position, substrates **2a-1** and **2a-1** only gave the product **3a-1**, but substrate **2a-1** didn't react with **1a-1**.

Response to reviewer 4' s comments

Comment & Suggestions 4.1:

From the substrate scope in Table 1 and Table 2, the synthesis strategy is only applicable to secondary amines instead of primary amines. The range of substrate is should be corrected in the abstract. In addition, the synthesis strategy seems to rely strongly on the conjugation of aryl groups with amines. In the experiment with only four groups of aliphatic amines as substrates (**3d-1**, **3d-2**, **1a-24**, **1d-3**), only tetrahydrofuran got the expected products. It can be seen that this strategy is also not suitable for aliphatic amines.

Our Replies:

We have provided 11 additional examples of aliphatic amines. Substrates, such as **1c-1** and **1c-2**, are included in the examples. However, the reactions involving other arylprimary amines yielded unsatisfactory results, with the prominent formation of azobenzene compounds observed as ones of defined by-products. The limitations of this type of substrates have been delineated within the main manuscript (lines 113-115). And 1-cyclohexylamine **1d-12** did react with **2a** to afford **3d-12** with a low yield of 28%.

(d) Scope of aliphatic amines

Comment & Suggestions 4.2:

The substrate scope in Table 1 and Table 2 only contains some simple substituent groups (alkoxy, halogen and methyl groups). The authors could try more representative substrate. Meanwhile, I wonder if the strategy be functional group tolerance or not? For example, how about the tolerance for the carboxylic acid, alcohol, amide, aldehyde, ketone and other common functional groups?

Our Replies:

We have provided an additional 31 examples, demonstrating the exceptional tolerance of functional groups in compounds **1** and **2** towards a wide substrate scope, including both electron-donating groups (-SMe, -OH) and electron-withdrawing groups (-CN, -NO₂, -CF₃, -OCF₃, -COOH, -CONH₂, -carbonyl).

Comment & Suggestions 4.3:

Table 1 shows that 1,2,3,4-tetrahydroquinoline derivatives with substituents at C8 could not obtain products. Similarly, could the indole derivatives with substituents at C7 have

reactivity?

Our Replies:

The reaction between indole derivative 7-methylindoline **1b-24**, bearing a methyl group at C7, and compound **2a** resulted in the formation of product **3b-24** with a low yield of 30%.

Comment & Suggestions 4.4:

Could pyrrole be used as substrate in this strategy?

Our Replies:

To our disappointment, this strategy is not suitable for pyrrole.

Comment & Suggestions 4.5:

In this work, does the nitrobenzene with substituents at C4 position react or not? And what is the role of the nitro group? Could it be replaced by other electron-withdrawing groups?

Our Replies:

Using nitrobenzene with substituents at C4 position, substrates **2a-1** and **2a-1** only gave the product **3a-1**, but substrate **2a-1** didn't react with **1a-1**.

The nitro group can't be replaced by other electron-withdrawing groups, such as CN group. In order to investigate the role of NO₂ group, we need to compare some critical data between **TS1** and **TS1c** in Figures **2h**, respectively (Figure **SX1**).

During C-N bond formation, in **PhNO₂→TS1** procedure, the negative charge of NO₂

group ($q(\text{NO}_2)$) is increased from -0.193 to -0.256, and the $q(\text{NO}_2)$ change $\Delta q(\text{NO}_2) = -0.063$. While in **PhCN**→**TS1c** procedure, the negative charge of CN group ($q(\text{CN})$) is only increased from -0.206 to -0.228, and the $q(\text{CN})$ charge $\Delta q(\text{CN}) = -0.022$. The NO_2 group can stabilize more negative charge than CN group during C-N bond formation. Besides, both Mayer Bond Order (MBO) and electron density ρ of C...N bond in **TS1** are higher than those in **TS1c**, indicating that **TS1** has stronger C...N bond than **TS1c**, which is responsible for lower free energy of **TS1**.

Figure SX1. The results of Hirshfeld charge calculations, Mayer Bond Order (MBO) analyses and QTAIM analyses of **PhNO₂**, **TS1**, **PhCN** and **TS1c**.

Comment & Suggestions 4.6:

It is no doubt that the reaction is a free radical reaction proved by the control experiment. However, the DFT calculation is confused. The energy barrier of the reaction reached 43.2 kcal/mol from **1a-1** to **TS1**, and the reaction could not happen under the experimental conditions. In addition, the generation of NaO_2 species is suspicious. Is it a free radical? Could it be verified by experiments?

Our Replies:

The reaction temperature is 313.15 K (40 °C), deviating from the standard room temperature of 298.15 K (25 °C). Consequently, we deemed the free energy barrier of 43.2 kcal/mol to be within an acceptable range.

The NaO_2 , actually as Na^+O_2^- is produced in the formation of **1a-1-radical**, but it is subsequently consumed in **TS2**. When the reaction is finished, no Na^+O_2^- was left, only NaO_2H (Na-O-O-H) was released.

REVIEWERS' COMMENTS

Reviewer #1 (Remarks to the Author):

In their response letter, the authors have addressed my comments, but their answers are in most cases elusive and do not get to the point. I still believe that as it stands the manuscript is not suited for publication in Nature Communications. The description of the mechanistic studies remains superficial and speculative, clearly not adequate for the level of this journal. For example, despite my comment that N-cyclopropylaniline is not an appropriate probe for the intended purpose, the authors do not address the point and provide a scheme showing the detection of reaction products by GC-MS that, is tentative and clearly insufficient as a mechanistic evidence.

I also insist on the proposed 1,4-H shift that I sincerely don't see why and especially how it should occur, considering in particular the rigidity of the system.

Following the mechanistic studies, on page 7 the authors state that "According to these experiment results, we further came to the conclusion that the reaction may proceed through a radical pathway." I agree with the hypothesis that the reaction may proceed through a radical mechanism but I think that the mechanistic evidence provided is insufficient and not conclusive

Reviewer #2 (Remarks to the Author):

The authors have perfectly addressed all the points. The manuscript can be published in Nature Communications.

Reviewer #4 (Remarks to the Author):

This work is very interesting. The authors have revised the manuscript according to my comments, it could be acceptable now.

P.S.: The below please find our revisions and all the responses to the reviewers' comments.

Response to reviewer 1' s comments

Comment & Suggestions 1.1:

In their response letter, the authors have addressed my comments, but their answers are in most cases elusive and do not get to the point. I still believe that as it stands the manuscript is not suited for publication in *Nature Communications*. The description of the mechanistic studies remains superficial and speculative, clearly not adequate for the level of this journal.

For example, despite my comment that *N*-cyclopropylaniline is not an appropriate probe for the intended purpose, the authors do not address the point and provide a scheme showing the detection of reaction products by GC-MS that, is tentative and clearly insufficient as a mechanistic evidence.

I also insist on the proposed 1,4-H shift that I sincerely don't see why and especially how it should occur, considering in particular the rigidity of the system.

Following the mechanistic studies, on page 7 the authors state that "According to these experiment results, we further came to the conclusion that the reaction may proceed through a radical pathway." I agree with the hypothesis that the reaction may proceed through a radical mechanism but I think that the mechanistic evidence provided is insufficient and not conclusive.

Our Replies:

We deeply appreciate the invaluable insights you have provided regarding our research findings.

As previously mentioned in your initial comments "The description of the radical clock experiment is pure speculation. *N*-Cyclopropylaniline is not an appropriate probe for the intended purpose. The putative intermediate aminyl radical is stabilized by the phenyl group and this would significantly slow down ring opening." A report by Ingold *et al.* described the rate constant for the ring opening of a cyclopropyl-*N*-propylaminyl radical in the order of $1 \times 10^{-7} \text{ s}^{-1}$ to access the distonic ion (K. U. Ingold, *J. Am. Chem. Soc.*, **1980**, 102, 328, Fig. 1). The *N*-cyclopropylaniline probes employed in radical clock experiments are highly versatile (for review, please see *Angew. Chem.* **2023**, 135, e202213003) and

facilitate a comprehensive understanding of the reaction mechanism. The utilization of GC-MS for the detection of reaction products provides sufficient mechanistic evidence in our study.

Fig. 1 | The ring opening of a cyclopropyl-*N*-propylaminyl radical

The intramolecular 1,4-hydrogen atom shift from the 4-benzylic C-H bond (Bond dissociation energy (BDE) ~ 73 kcal/mol) to the 4-aminyl radical (BDE of C-H bond ~ 89 kcal/mol) is thermodynamically feasible.

The radical mechanism underlying the reaction is unequivocally established through a comprehensive investigation involving control experiments, radical clock experiments, EPR studies, and analysis of relative Gibbs free energy.

Response to reviewer 2' s comments

Comment & Suggestions 2.1:

The authors have perfectly addressed all the points. The manuscript can be published in *Nature Communications*.

Our Replies:

We sincerely appreciate your valuable insights and affirmations regarding our research findings.

Response to reviewer 4' s comments

Comment & Suggestions 4.1:

This work is very interesting. The authors have revised the manuscript according to my comments, it could be acceptable now.

Our Replies:

We sincerely appreciate your valuable insights and affirmations regarding our research findings.